⊙ | Open Peer Review | Physiology and Metabolism | Research Article

# The potential mechanisms of reciprocal regulation of gut microbiota-liver immune signaling in metabolic dysfunction-associated steatohepatitis revealed in multi-omics analysis

Zhaoyang Lu,[1] Ligen Yu,[1,2] Yun Bai,[3] Yifeng Cui,[1] Meixin Shi,[2] Zhitao Li,[2] Xiaoxue Li,[2] Xin Zhong,[2] Ye Jin,[1] Can Wei[1,2]

**ABSTRACT**  As a commonly known aggressive liver-related manifestation within the spectrum of metabolic syndrome with a significant risk of progressing to cirrhosis and hepatocellular carcinoma, metabolic dysfunction-associated steatohepatitis (MASH) is closely intertwined with obesity, insulin resistance, and dyslipidemia. Although the gut microbiota is implicated in MASH progression, the underlying mechanisms require further investigation. In this study, we sought to combine the analysis of the liver transcriptome, circulating metabolome, and gut microbiota to investigate the potential molecular mechanisms underlying the reciprocal regulation between gut microbiota and liver immune signaling. We utilized a high-fat and methionine/choline-deficient diet (HFMCD)-induced MASH model in a db/db mouse. Following annotation analysis using KEGG and Metorigin, a comprehensive correlation analysis was conducted among these genes and specific metabolites (such as L-glutamine, isocitric acid, putrescine, pyroglutamic acid, rhamnose) and gut microbiota genera (*Enteroccus* and *Romboutsia*). The results revealed intricate interactions among the liver's immune microenvironment, the metabolome, and the gut microbiota. These interactions suggest a potential regulatory mechanism for metabolic disorders and immune responses.

**IMPORTANCE**  Our multi-omics analysis showed that the interactions between gut microbiota and liver immune responses mediated by the disorders in lipid, amino acid, and glucose metabolism are associated with activation of the JAK-STAT and NF-κB signaling pathway in MASH. The multi-omics analysis provides valuable insights into the interactions among microbiota, circulating metabolites, and immune signaling. These insights can be harnessed to enhance the management of MASH.

**KEYWORDS**  MASH, multi-omics analysis, transcriptome, metabolome, gut microbiota

Metabolic dysfunction-associated steatohepatitis (MASH), representing the advanced phase of metabolic dysfunction-associated fatty liver disease (MAFLD), has become an ever-more-common liver disorder. It is intricately tied to the global upsurge in obesity and metabolic syndrome (1). Previously known as non-alcoholic steatohepatitis or NASH, MASH received its new nomenclature when professional societies made the change last year (2, 3). Nowadays, therapeutic strategies primarily center on dietary adjustments and pharmacological interventions (4, 5). Regrettably, these approaches often have limited efficacy and fail to fully tackle the fundamental pathophysiological processes. As a result, there is a pressing need for in-depth research to clarify the molecular mechanisms underpinning MASH. Such knowledge could pave the way for novel therapeutic approaches and improve patient prognoses.

The gut microbiota, functioning as a complex and dynamic ecosystem (6), interacts with the liver via the gut-liver axis, which delineates bidirectional regulation of

**Peer Reviewers** Hongzhu Li, Xiamen University, Xiamen, Fujian, China; Yuehong Wang, Renji Hospital affiliated to Shanghai Jiaotong University, Shanghai, China

Address correspondence to Ye Jin, hydjinye@hotmail.com, or Can Wei, canwei528@hrbmu.edu.cn.

Zhaoyang Lu, Ligen Yu, Yun Bai, and Yifeng Cui contributed equally to this article. The order of the first authors reflects the amount of time and effort each author devoted to this project.

The authors declare no conflict of interest.

metabolites, immune cells, and cytokines, alongside metabolic and immune interdependencies between these two vital organs (7). In a state of physiological equilibrium, the various microbial communities within the gut maintain a stable balance in their compositional proportions. Conversely, when dysbiosis occurs, it disrupts the structural integrity of the intestine, disturbs functional homeostasis, and reduces microbial diversity. Such disruptions can potentially instigate inflammation in host tissues, metabolic imbalances, and immune dysregulation (8). In fact, the mutual dependence and intricate communication between the liver and gut, especially those mediated by the gut microbiota, have been demonstrated to be closely associated with the metabolic dysfunction and inflammation that are hallmarks of MASH (3, 9, 10). Studies have indicated that the gut microbiota contributes to the development of MASH by undermining the integrity of the intestinal barrier and boosting the production of harmful metabolites (11, 12). For instance, during the progression of high-cholesterol diet-induced MASH-HCC, microbial dysbiosis manifests as increased relative abundances of *Mucispirillum*, *Desulfovibrio*, *Anaerotruncus,* and *Desulfovibrionaceae* alongside decreased levels of *Bifidobacterium* and *Bacteroides*. In addition, there are alterations in the levels of bacterial metabolites such as elevated taurocholic acid and reduced 3-indole-propionic acid (11). Further investigations have shown that when dysbiotic microbiota is transplanted into germ-free mice, it can exacerbate hepatic lipid accumulation and inflammation, while atorvastatin can partially restore the microbial structure and prevent the development of hepatocellular carcinoma (11). Nevertheless, the exact mechanisms by which the microbiota regulates the progression of MASH remain largely unknown. This clearly emphasizes the need for comprehensive analyses that integrate transcriptomic, metabolomic, and microbiome data sets to better understand the complex interactions between the host, metabolites, and microbiota.

Previous research has firmly established a robust link between dietary factors and the progression of metabolic diseases, including fatty liver diseases (13). Significantly, studies have shown that certain dietary patterns can exacerbate liver conditions, highlighting the pivotal role of nutrition in the phenotypic manifestations of MASH (14). The use of animal models, especially db/db mice, has been crucial in elucidating these relationships (15). When these mice are fed a high-fat diet lacking methionine and choline, it offers valuable insights into the phenotypic manifestations of MASH. This, in turn, paves the way for a more in-depth exploration of this disease in the context of dietary impacts.

The bile acid enterohepatic circulation between the gut and liver plays a critical role in maintaining host metabolic homeostasis (16). As the co-metabolites derived from host synthesis and microbial modification, bile acids can directly regulate the metabolic-dysfunction-associated steatohepatitis (MASH) through activating nuclear receptors including farnesoid X receptor (*FXR*) and Takeda G protein-coupled receptor 5 (*TGR5*), while indirectly shaping gut microbiota composition via modulating the intestinal epithelial microenvironment (17–20). Clinical studies have demonstrated that during the progression of metabolic dysfunction-associated fatty liver disease (MAFLD), bile acid levels increase (21), and this phenomenon is closely linked to the bile acid dysregulation induced by obesity- and type 2 diabetes (T2D) (22, 23). Such metabolic disturbances drive structural remodeling of the gut microbiota, characterized by decreased Bacteroidetes and increased Firmicutes abundance, a pattern consistently observed in high-fat diet-fed db/db mouse models.

In this study, we utilized multi-omics analyses, which integrated microbiome, metabolome, and transcriptome profiling, to comprehensively elucidate the complex biological mechanisms underlying the pathogenesis of MASH. First, we identified the differentially expressed genes and metabolites between the MASH and the Control Groups, and characterized their functional roles in disease progression. Subsequently, we analyzed gut microbial diversity and taxonomic composition via 16S rRNA sequencing (MASH vs. Control Groups). Following KEGG pathway annotation and Metorigin-based metabolite tracing, we mapped the co-regulated metabolic pathways and microbial-derived metabolites. Through integrative correlation analyses that linked

gene expression patterns, gut microbiota dynamics, and serum metabolite profiles, we delineated the key gene-microbe-metabolite associations that drive MASH pathology. In summary, by employing multi-omics analysis, this study endeavors to offer novel insights into the gut-liver axis and pave the way for innovative treatment strategies that could substantially enhance patient outcomes.

## RESULTS

### Transcriptomic analysis

To explore the pathogenesis of MASH in db/db mice fed with a high-fat, methionine- and choline-deficient diet, RNA sequencing was performed to identify differential gene expression profiles between the Control and MASH Groups. We identified 619 differentially expressed genes (DEGs), including 447 upregulated genes and 172 downregulated genes (Fig. 1A). A heatmap visualizing the expression abundance of DEG is presented (Fig. 1C). A Venn diagram analysis revealed 11,372 shared genes, with 962 genes exclusive to the MASH Group and 256 exclusive to the Control Group (Fig. 1B). Gene Ontology biological process (GO-BP) enrichment analysis revealed significant enrichment

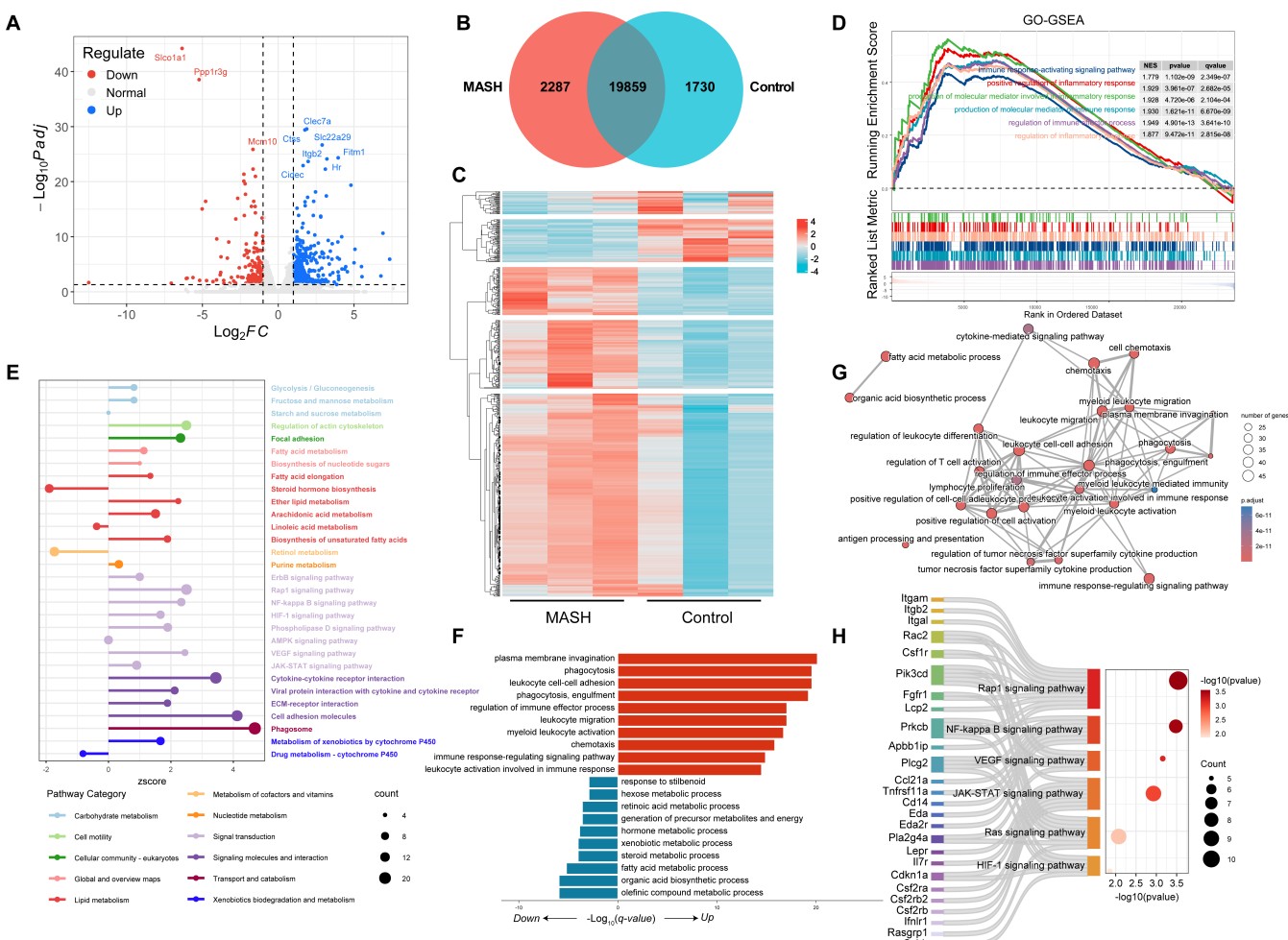

FIG 1 The altered transcriptome and immune signaling in the liver ($n = 3$). (A) Volcanic plot of DEGs between the MASH and Control Groups. (B) Venn diagram constructed from the identified genes. (C) Heatmap displaying the expression abundance of DEGs. (D) GSEA plot for inflammation- and immune-related GO_BP terms between the MASH and the Control Groups. (E) Lollipop plots representing the top 30 significant pathways, excluding those within the Human Diseases and Organismal Systems category, based on the KEGG database. (F) Pathway enrichment analysis of overlapping downregulated (reference threshold: fold change, <0.5; $P$.adjust value, <0.05) and upregulated (reference threshold; fold change, >2; $P$.adjust value <0.05) genes was performed based on the GO database. (G) Network diagram of the top 25 GO_BP terms. (H) KEGG enrichment sankey bubble plots for the selected gene sets.

and upregulation of several immune and inflammatory processes (Table S1). The top three enriched GO-BP terms included the regulation of immune effector process, the production of molecular mediators of immune response, and the immune response-activating signaling pathway; the top three inflammatory terms were the regulation of inflammatory response, the positive regulation of inflammatory response, and the production of molecular mediators involved in the inflammatory response (Fig. 1D). The pathway enrichment analysis using the KEGG database revealed that the DEGs were associated with key metabolic pathways, such as glycolysis, fatty acid metabolism, and xenobiotic metabolism via cytochrome P450. Furthermore, these genes were involved in signaling pathways like NF-κB signaling and cytokine-cytokine receptor interaction (Fig. 1E). We conducted a GO-BP enrichment analysis on the DEGs shared between the two groups using GO_BP enrichment analysis. The upregulated genes were primarily enriched in immune pathways, including the immune response, phagocytosis, and leukocyte activation and adhesion. The downregulated genes were predominantly enriched in pathways related to metabolic dysfunction, such as fatty acid metabolism, steroid metabolism, and organic acid biosynthesis (Fig. 1F). Among the top most enriched BPs assessed, we found that 23 were closely connected to immunity (Fig. 1G). Based on a gene set consisting of 23 BPs, the five signaling pathways that were significant in the KEGG enrichment results play a crucial role in these biological processes (Fig. 1H).

## Metabolomics analysis

A total of 10,172 peaks and 992 compounds were identified in the liver metabolome. The score plots of component analysis, partial least squares discriminant analysis, and orthogonal partial least squares discriminant analysis score plots are presented in Fig. 2A through C. When compared to the Control Group, the MASH Group exhibited 177 downregulated and 90 upregulated metabolites (Fig. 2D). These differentially expressed metabolites mainly fell into the functional categories of lipids and lipid-like molecules, organic acids and their derivatives, organoheterocyclic compounds, and organic oxygen compounds (Fig. 2E). Analysis of the metabolome map identified revealed several enriched metabolic pathways such as tryptophan metabolism, linoleic acid metabolism, primary bile acid biosynthesis, glyoxylate and dicarboxylate metabolism, and arginine biosynthesis (Fig. 2F).

To explore the intricate relationships between the genes and metabolites more comprehensively, we integrated the transcriptome and metabolome using KEGG enrichment analysis. This integrative approach shared 130 shared KEGG pathways, including linoleic acid metabolism, biosynthesis of unsaturated fatty acids, and fructose and mannose metabolism (Fig. 2G). Subsequently, a Spearman correlation network analysis was performed on 44 differentially expressed metabolites (DEMs) and 46 DEGs derived from 18 KEGG pathways (Table S2). This analysis uncovered 249 significant metabolite-gene correlations, involving 43 DEMs and 44 DEGs (Fig. 2H and I). In particular, APC, PC(14:0/22:2[13Z,16Z]), guanidoacetic acid, traumatin, and linoleic acid emerged as the top five key core metabolites (with a degree of ≥13), showing a high degree of association with most DEGs. Meanwhile, cytochrome P450 family 2 subfamily C member 2 (*Cyp2c2*), cytochrome P450 family 3 subfamily A member 11 (*Cyp3a11*), hydroxyacid oxidase 2 (*Hao2*), acetyl-CoA acyltransferase 1B (*Acaa1b*), and aldo-keto reductase family 1 member B7 (*Akr1b7*) were identified as the top five core genes (with a degree of >8) highly associated with most DMs. In conclusion, these comprehensive analyses indicate that metabolites and metabolism-related genes, which are enriched in 18 shared pathways, might be central to the molecular connection underlying MASH.

## Gut microbiota analysis

A comparative analysis of gut microbiota diversity at the genus level (Fig. 3A and B) revealed a notable difference in the Shannon index between the MASH and Control Groups, while the Simpson index showed no significant difference. This indicates that

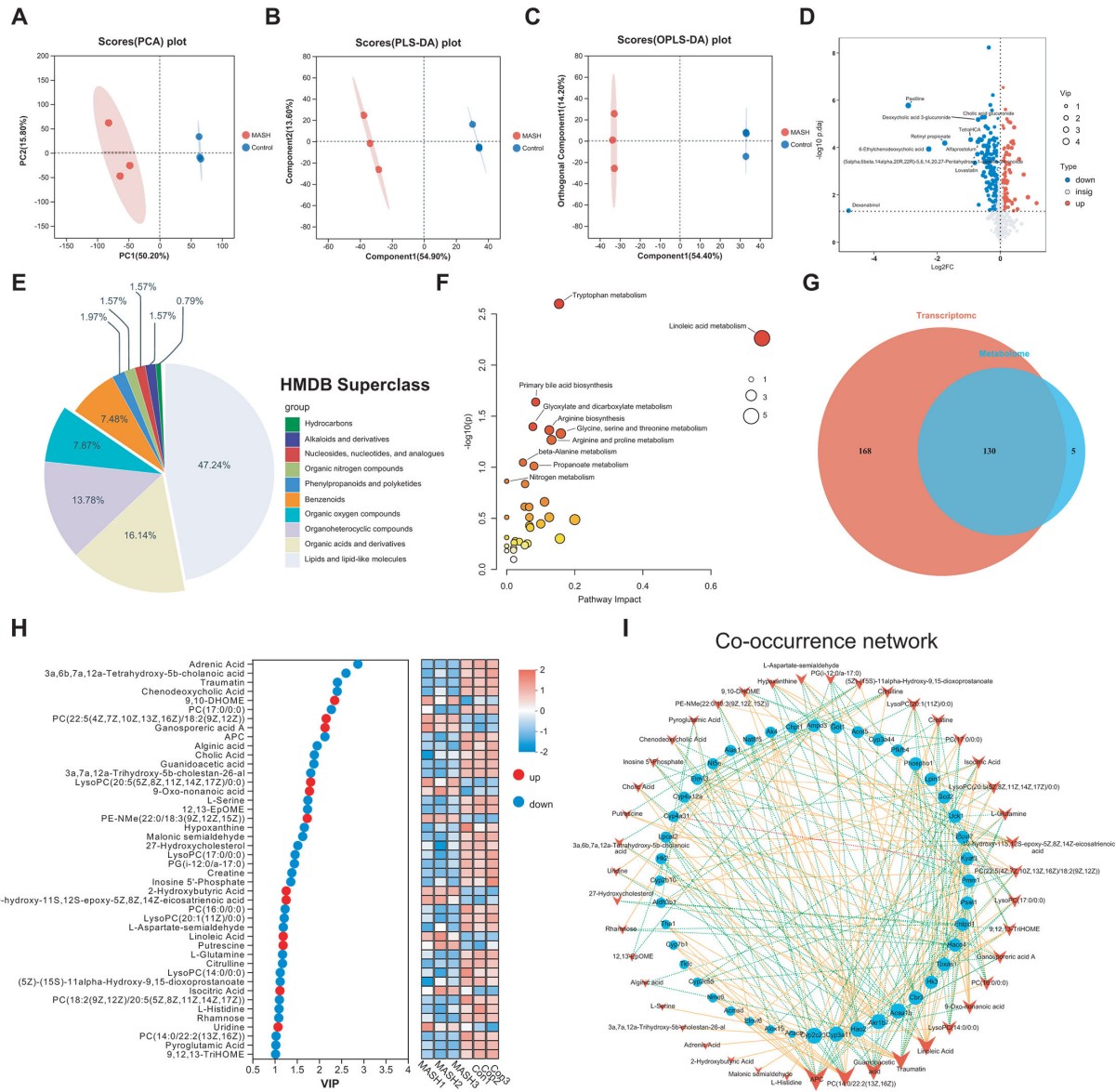

**FIG 2** The aberrant metabolome of MASH and its interactions with co-metabolism genes ($n = 3$). (A) The principal component analysis was used to illustrate the differences of metabolites between the two groups: (B) partial least squares discriminant analysis and (C) orthogonal partial least squares discriminant analysis score plots. (D) A volcanic plot shows the DMs between the MASH and the Control groups. (E) A pie plot of the HMDB Superclass of DMs between the two groups is provided. (F) The KEGG terms of DMs from enrichment analysis are depicted. (G) A Venn diagram demonstrates the correlation between transcriptomics and metabolomics pathways. (H) A heatmap shows the relative expression abundance and VIP of 44 metabolites. (I) The associations between liver differentially expressed genes and serum differential metabolites within 18 shared pathways are presented. Each co-occurring pair between two items had an absolute Spearman correlation coefficient greater than 0.92 (orange solid line for positive correlation ($r \geq 0.92$); green dashed line for negative correlation ($r \leq -0.92$)), with $P$-values below 0.01. The size of the nodes is proportional to their connectivity with other nodes in the network.

the two groups display substantial differences in species richness and evenness at the genus level, yet there is no significant disparity in the relative abundance of dominant genera. Principal component analysis (PCA) showed a distinct separation between the gut microbiota of the MASH and the Control Groups, both at the genus and species levels (Fig. 3C and D). We analyzed 16S rRNA gene sequencing data to discern differences in gut microbial composition. In total, 579 operational taxonomic units (OTUs) were identified, comprising 121 genera and 180 species, the Venn plots showed marked differences between the two groups at both the genus and species levels (Fig. 3E and F).

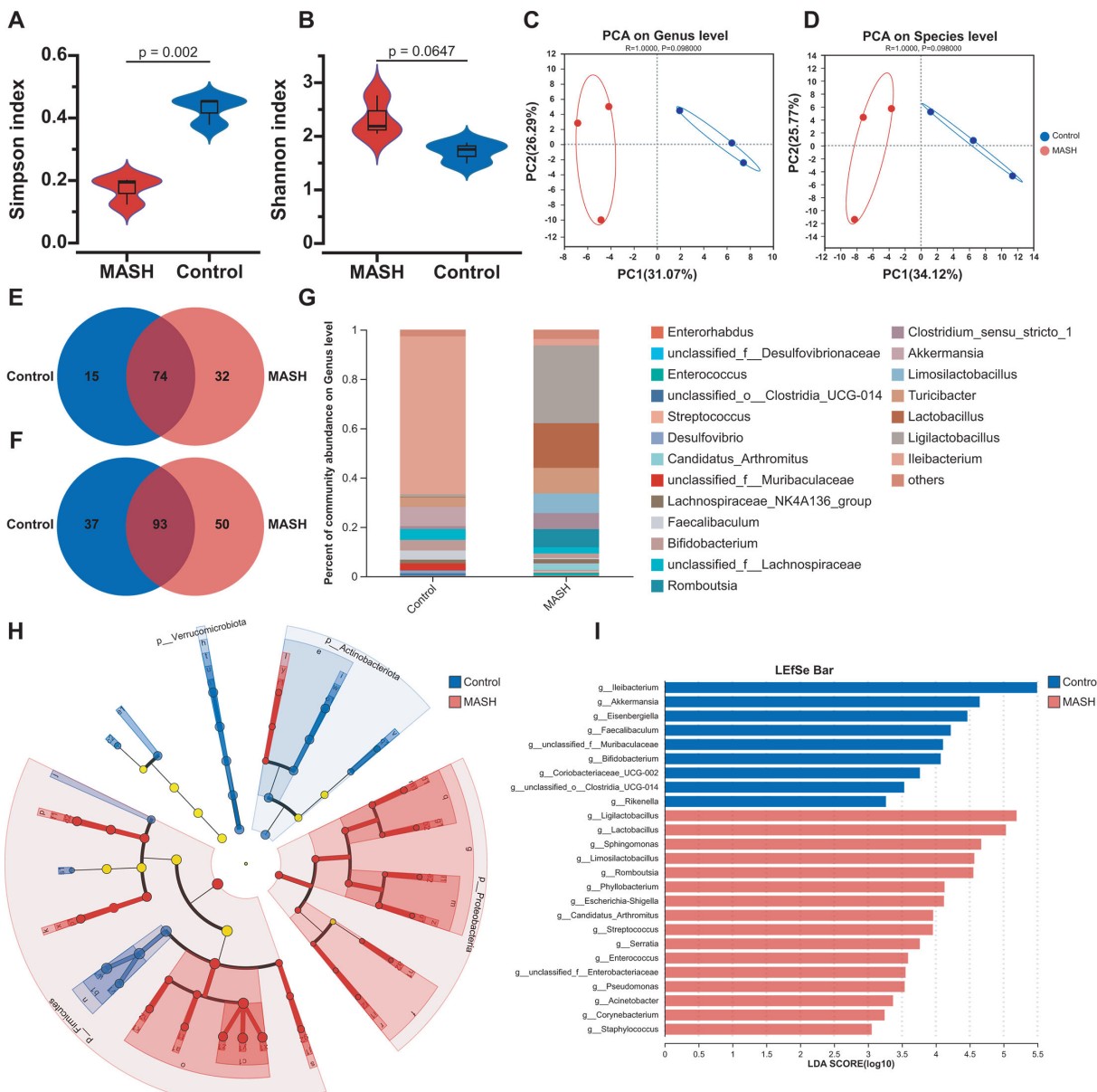

**FIG 3** Characterization of gut microbiota in the control and MASH groups (*n* = 3). (A) The differences in alpha diversity between the two groups are shown by the Shannon index (Student's *t*-test) and (B) the Simpson index (Student's *t*-test). (C) Multiple-sample principal component analysis (PCA) was performed at the genus level and (D) the species level. (E) Venn diagrams show the relationships between the two groups at the genus level and (F) the species levels. (G) Stacked bar graphs depict the relative abundance of the top 20 genera. (H) The LEfSe was used to identify differences in community composition between the two groups. In the figure, nodes of different colours represent significant effects on the differences between the groups. (I) The column chart displays biomarkers for genera with an at LDA score of >3.

Histograms (Fig. 3G) illustrate the gut microbiota community structure and the relative abundance of major genera and species. At the genus level, the dominant genera were *Ileibacterium*, *Ligilactobacillus*, *Lactobacillus*, *Turicibacter*, and *Limosilactobacillus* (Fig. 3G). At the species level, the most abundant species included *Ileibacterium valens*, *Lactobacillus murinus*, uncultured *Turicibacter*, unclassified *Lactobacillus*, and *Lactobacillus reuteri* (Fig. S1). Linear discriminant analysis effect size (LEfSe) analysis (with an LDA threshold of >3) identified 25 genus-level features that distinguished the Control and the MASH Groups. Specifically, the Control Group was enriched in 16 genera, including *Ligilactobacillus*, *Lactobacillus*, *Sphingomonas*, and *Limosilactobacillus*, while the MASH Group was

enriched in nine genera, including *Ileibacterium*, *Akkermansia*, and *Eisenbergiella* (Fig. 3H and I).

## Integrative analysis of gut microbiota and serum metabolome

We further explored the potential associations between gut microbiota and serum metabolites. Through annotation to MetOrigin, the sources of the 267 differential metabolites, were classified as follows: host-derived ($n = 50$), microbiota-derived ($n = 60$), resulting from host-microbiota co-metabolism ($n = 4$), and others with heterogeneous origins ($n = 196$) (Fig. 4A and B). Based on the KEGG database, PICRUST2 analysis predicted the top 150 most abundant functions within the gut microbial community. These functions encompassed the biosynthesis of secondary metabolites, microbial metabolism in diverse environments, amino acid biosynthesis, ribosome-related functions, and carbon metabolism. A heatmap (Fig. 4C) was crafted to represent these

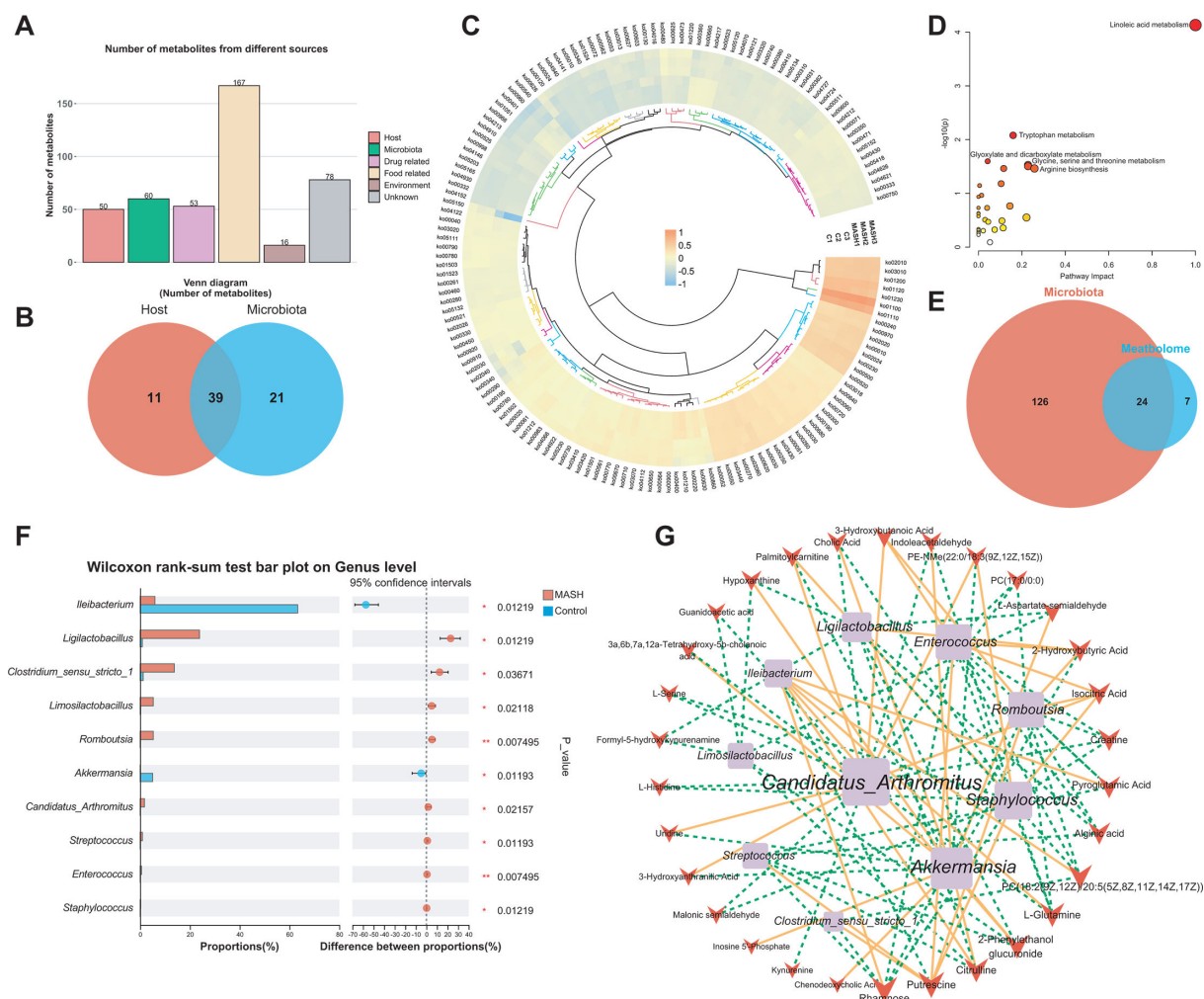

**FIG 4** The aberrant microbiome of MASH and its interactions with potential gut microbiota metabolites. (A) Barplots were utilized to illustrate the potential origins of differential metabolites, along with the corresponding number of metabolites from each source. (B) A Venn diagram was employed to demonstrate the correlation between host and microbiota metabolites. (C) A circular heatmap was generated to display the predicted metabolic pathways with different abundances between groups. (D) A scatterplot was presented to show metabolite enrichment pathways of potential gut microbiota. (E) A Venn diagram was constructed to represent the interrelationship of the top 150 enriched predicted metabolic pathways with potential gut microbiota metabolites. (F) Column graphs were created to depict gut microbial differences at the genus level based on Wilcoxon's rank sum test. (G) The associations between gut differential genera and serum differential metabolites in 24 shared pathways were analyzed. Each co-occurring pair between two items had an absolute Spearman correlation above 0.81 (orange solid line for positive correlation ($r \geq 0.81$); green dashed line for negative correlation ($r \leq -0.81$)), with *P*-values below 0.05. Additionally, the size of nodes was proportional to the degree of connectivity with other nodes in the network.

top 150 pathways effectively visualizing their relative abundances (Fig. 4C). Subsequently, an enrichment analysis of the 60 microbiota-associated metabolites identified 31 enriched pathways, 24 of which overlapped with those predicted by PICRUSt2 (Fig. 4D and E; Table S3). In line with the Wilcoxon's rank sum test at the genus level, the relative abundances of 10 genera exhibited significant differences between the MASH and the Control Groups. In the MASH Group, the abundance of *Ileibacterium* and *Akkermansia* decreased, while those of *Ligilactobacillus*, *Clostridium sensu stricto 1*, and *Limosilactobacillus* increased (Fig. 4F).

Subsequently, we selected all 30 metabolites from the 24 shared pathways, and 10 genera from the genus-level difference analysis (Fig. 4F) were selected for subsequent correlation analyses. Spearman correlation analysis revealed significant associations between the 10 genera and 30 metabolites (Fig. 2G). *Candidatus* Arthromitus, *Akkermansia,* and *Staphylococcus* were identified as the three key genera most strongly associated with most metabolites. Meanwhile, rhamnose, putrescine, citrulline, 2-phenylethanol glucuronide, and L-glutamine were the five key metabolites most strongly associated with most genera.

## The putative mechanistic correlations among serum metabolites, gut microbiota, and liver transcriptome

To gain a more in-depth understanding of the core interactions between liver genes, serum metabolites, and gut microbiota in MASH, Mantel, and Spearman correlation analyses were performed on metabolism-related genes, JAK-STAT and NF-κB signaling pathway genes, 44 significantly altered metabolites, and 10 differentially abundant genera (Fig. 5A and B; Fig. S2). The results validated that the distance-corrected dissimilarities of genes and genera were initially correlated with those of metabolites. Overall, 17 metabolites exhibited the strongest correlations with 10 different genera and metabolism-related genes, including PC(17:0/0:0), LysoPC(20:1[11Z]/0:0), L-glutamine, rhamnose, and creatine. Additionally, nine metabolites exhibited the most significant correlation with 10 different genera and genes in the JAK-STAT and NF-kappa B signaling pathway, including rhamnose, PC(17:0/0:0), creatine, and L-glutamine. Specifically, a dot-plot diagrams was employed to illustrate the correlation coefficients between liver genes and gut microbiota (Fig. 5C and D). Based on the 17 and 9 that best explained the relationships in each of the two Mantel analyses described above, network diagrams were constructed to illustrate the one-to-one interactions between genes in the JAK-STAT and NF-κB signaling pathways and metabolites, as well as between gut microbiota and metabolites (Fig. 5E and F).

## DISCUSSION

As a critical global public health challenge, metabolic dysfunction-associated steatohepatitis (MASH) has affected approximately 5% of the adult population worldwide and served as a leading driver of liver failure, thereby fueling the increasing demand for liver transplantation. The pathological progression of MASH originates from the aberrant accumulation of lipotoxic species in the liver, thereby triggering sustained inflammatory responses and hepatocyte damage. As the disease progresses, the inflammatory microenvironment promotes the deposition of pathological fibrotic tissue, ultimately resulting in a progressive decline in liver function. It is closely linked to systemic metabolic disorders and involves complex interactions between the liver's immune environment and the gut microbiota (24–26). Previous studies have shown that innate immunity is sufficient to cause MASH, and adaptive immunity is necessary for the development and progression of MASH (27). Furthermore, through inflammatory communication between innate and adaptive immunity, cells of adaptive immunity activate and instruct the function of innate immune responses (28–30). Our results suggest that in MASH, the activation of the JAK-STAT and NF-κB signaling pathways is closely associated with cytokine-related biological processes. As cytokine-related

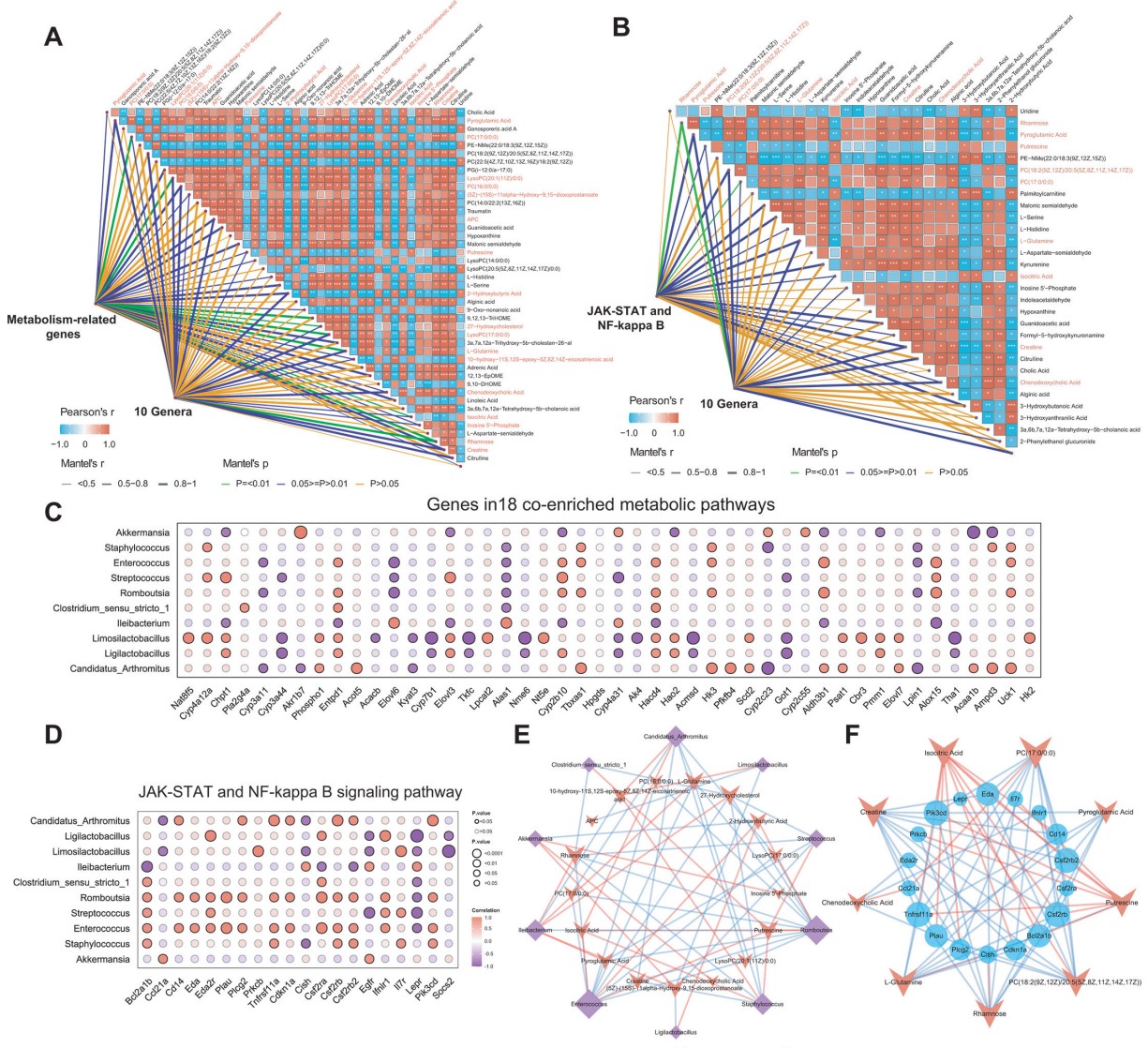

**FIG 5** The variations in gut microbiota genera and serum metabolites exhibit pronounced associations with genes expression of the JAK-STAT and NF-κB signaling pathways, as well as metabolism-related genes expression. (A) Through Mantel analysis, both 46 metabolism-related genes and 10 differential genera were linked to each of the 44 metabolites, respectively. Among these, 17 of these metabolites were significantly associated with metabolism-related genes and 10 differential genera (Mantel's $P < 0.05$, and Mantel's $r > 0.5$). (B) Using Mantel analysis, both 20 genes of the JAK-STAT and NF-κB signaling pathways and 10 differential genera were each linked to 30 metabolites. Among these, nine metabolites were significantly associated with the genes and 10 differential genera (Mantel's $P < 0.05$, and Mantel's $r > 0.5$). (C) The correlation between the 10 genera and genes expression in 18 co-enriched metabolic pathways was investigated. (D) The correlation between the 10 genera and the genes expression of the JAK-STAT and NF-κB signaling pathway was explored. (E) The correlation between the 10 genera and the 17 serum metabolites was analyzed. (F) The correlation between the liver genes of the JAK-STAT and NF-κB signaling pathways and the nine serum metabolites was examined. The red line indicates a positive correlation ($r \geq 0.81$); the blue line indicates a negative correlation ($r \leq -0.81$), and the size of nodes was proportional to the degree of connectivity with other nodes in the network.

signaling pathways, these two pathways, as cytokine-associated signaling pathways, may play a crucial role in the occurrence and development of MASH.

Likely, the upregulation and downregulation of the 619 DEGs are potentially involved in the occurrence, progression, and persistence of the disease. For example, upregulated genes may include those that promote inflammatory responses, immune cell activation, and cytokine release (31–33). All these are critical processes in the pathophysiology of MASH. Downregulated genes may include those involved in fatty acid metabolism,

cholesterol synthesis, and antioxidant defense (34–36), which contribute to liver injury by worsening metabolic disorders and oxidative stress.

The analysis of metabolic pathways supplements this by revealing significant impacts on glycolysis, fatty acid metabolism, and cytochrome P450-mediated exogenous metabolic pathways. These results indicate the existence of a feedback loop between metabolic disorders and inflammation. Specifically, the abnormal accumulation of metabolic substrates and metabolites affects liver resident immune cells and liver sinusoidal endothelial cells, giving rise to persistent inflammation, which, in turn, accelerates liver injury and fibrosis (37).

Integrating metabolomic and transcriptomic data offers new insights into the mechanisms underlying MASH and may lead to targeted therapeutic strategies. Our metabolomic analysis identified various altered metabolites in MASH, mainly including lipids, organic acids, heterocyclic compounds, and oxygen-containing compounds. This supports earlier findings that emphasize the crucial role of lipid imbalance in the progression from simple steatosis to steatohepatitis (38, 39). Our findings indicated significant changes in the metabolism of tryptophan and linoleic acid metabolism, along with disruptions in the metabolic pathways of primary bile acid biosynthesis, as well as the glyoxylate/dicarboxylate and arginine metabolic pathways. These pathways are recognized as key factors affecting liver inflammation, oxidative stress, and cell death, all of which contribute to the occurrence of MASH (40–42).

Through the KEGG enrichment analysis, we combined transcriptomic and metabolomic data to identify 18 shared metabolic pathways that reveal the complex interactions between genes and metabolites. Our correlation network analysis identified critical metabolic hubs (such as APC, PC, guanidinoacetic acid, atin, and linoleic acid) and key genetic hubs (including Cyp2c2, Cyp3a11, Hao2, Acaa1b, and Akr1b7). Genetic changes in the liver, such as epigenetic variations, may lead to complex metabolic diseases (24). Metabolite-mediated epigenetic changes represent an adaptive mechanism of cell signaling. The recently discovered β-hydroxybutyrate-mediated epigenetic pathway, namely, lysine β-hydroxybutyrate (Kbhb), links metabolism to gene expression (42). Mass spectrometry analysis of β-hydroxybutyrylomics in the mouse liver revealed the presence of Kbhb sites in proteins enriched in fatty acid, amino acid, detoxification, and one-carbon metabolism pathways (40). These pathways are closely associated with MASH, suggesting that Kbhb may play an important role in theliver metabolism in MASH.

The findings of this study indicate that specific alterations in the microbiome-gut-liver axis trigger metabolic disturbances and immune responses, suggesting that a healthy gut microbiota is essential for the development and progression of MASH. However, there are still limits to studies investigating the role and potential molecular mechanisms of changes in the MASH gut microbiome. Therefore, this study focuses on the role and potential molecular mechanisms of gut microbiome changes in the MASH. The most significant changes were observed in the increased abundance of *Erysipelotrichaceae* and *Lactobacillaceae* families. *Bacteria* from the families *Erysipelotrichaceae* and *Lactobacillaceae* (phylum *Firmicutes*) dominate the gut microbiota of mice and have also been detected in the gut of other animals (43, 44). In this study, the lack of significant differences in Simpson index values, combined with the analysis of microbiota variation at the genus level, suggests two plausible explanations. First, the relative abundance of the most dominant genera did not change significantly, such as *Lactobacillus*, *Turicibacter*, *Akkermansia*, unclassified f *Lachnospiraceae, Bifidobacterium,* and *Faecalibaculum*. Second, the genera with significant inter-group differences occupied a relatively minor proportion within the microbial community, such as *Staphylococcus*, *Enterococcus*, *Streptococcus,* and *Candidatus* Arthromitus.

In this study, metabolic enrichment analysis of db/db mice fed a high-fat methionine-deficient diet suggested disorders in lipid metabolism, amino acid metabolism, and bile acid metabolism, which is consistent with findings in humans (16, 45, 46). MASH is associated with the modulation of the host's response to the intestinal microbiota. This modulation functions, in part, through microbiome-related serum metabolites and

is further associated with liver metabolic disorders and immune responses in MASH. Comparative analysis revealed significant differences in microbial predicted functions annotated to KEGG pathways, particularly in glucose metabolism, glycerolipid metabolism, secondary metabolite biosynthesis, and amino acid biosynthesis, all of which serve as evidence for this perspective (Fig. S3).

We found evidence that L-glutamine, isocitric acid, putrescine, pyroglutamic acid, and rhamnose may act as potential messenger molecules in the gut-liver axis, and these findings suggest that gut-associated microbiota and metabolites, particularly putrescine and rhamnose, are implicated in the progression of MASH. Mechanistic studies reveal that rhamnose directly binds to and activates solute carrier family 12 member 4 (*SLC12A4*) on macrophages, triggering the Ras-related C3 botulinum toxin substrate 1 (*Rac1*)/cell division cycle 42 homolog (*Cdc42*) signaling axis to enhance phagocytic activity and reduce sepsis-induced organ damage and mortality [47]). Notably, the level of rhamnose is negatively correlated. rhamnose levels inversely correlate with the abundances of *Romboutsia* and *Enterococcus* genera, while the enrichment of these genera in the liver and adipose tissue is positively correlated, and hepatic and adipose tissue enrichment of these genera positively associates with the severity of metabolic dysfunction-associated fatty liver disease (MAFLD) (48). Furthermore, the negative correlation between rhamnose and genes in the hepatic JAK-STAT/NF-κB pathway genes suggests its immunomodulatory role through host-microbial interactions. By contrast, microbially derived putrescine regulates intestinal immune homeostasis via the polyamine metabolic pathway. Elevated putrescine levels of putrescine promote colonic epithelial proliferation and macrophage differentiation (49–51). Unlike rhamnose, putrescine exhibits positive correlations with hepatic JAK-STAT/NF-κB signaling genes, indicating its involvement in the immune responses during the progression of MASH progression. Collectively, these findings highlight the bidirectional regulatory roles of gut microbial metabolites in shaping the immune-metabolic crosstalk within the gut-liver axis. A limitation of this multi-omics study lies in the sample size of mice in each group ($n$ = 3), which may reduce the statistical power to resolve subtle gut microbiota-liver immune interactions in MASH. However, the identified reciprocal regulatory signatures, such as putrescine and rhamnose-mediated microbiota-immune crosstalk, demonstrated high consistency across all samples (correlation $r$ > 0.81 among genes in the JAK-STAT and NF-κB signaling pathways, putrescine and rhamnose, and *Enterococcus* and *Romboutsia*; Fig. 5), and align with prior mechanistic reports linking gut-derived metabolites to MASH progression.

In summary, these metabolites show significant correlations with differential genera and metabolic genes within these pathways, indicating their potential roles as key messengers mediating the gut-liver axis interactions. Additionally, the role of gut microbes, especially the genera of *Enterococcus* and *Romboutsia*, in modulating the gene expression of the JAK-STAT and NF-κB signaling pathway, as well as the levels of five key metabolites, underscores the potential for microbe-based therapeutic approaches. Probiotic or prebiotic interventions that promote beneficial gut bacteria can be investigated as a means to support the treatment of MASH through the gut-liver axis.

## Conclusions

To date, our understanding of the interconnections and causal relationships between gut microbes and the chronic inflammatory and immune responses in MASH remains limited. The elucidation of potential regulatory mechanisms has spurred the development of prevention strategies and therapeutic approaches. Our multi-omics analysis showed that the interactions between the gut microbiota and liver immune responses mediated by the disorders in lipid, amino acid, and glucose glucose metabolism were associated with activation of the JAK-STAT and NF-κB signaling pathway in MASH induced by an HFMCD in db/db mice. Meanwhile, metabolites closely related to the transcriptome and gut microbiota, especially L-glutamine, isocitric acid, putrescine, pyroglutamic acid, and rhamnose, are potential messengers of the enterohepatic axis. At the same time,

the genera *Enterococcus* and *Romboutsia* are crucial players in the coordination of serum metabolome, gut microbiota, and immune genes, and may trigger the host's inflammatory immune responses through the microbiota-liver axis. Despite the paucity of evidence for direct causality, our findings also suggest that efforts to improve gut ecology may be a promising approach for the treatment of MASH.

## MATERIALS AND METHODS

### Animal model and ethical considerations

We utilized male db/db mice (on a C57BL/6J background), known for their genetic susceptibility to obesity, insulin resistance, and hyperglycemia. Referring to a previous report (52–54), the mice were obtained from a commercial supplier and housed in a controlled environment with a 12 hour light/dark cycle at a temperature of 22°C ± 2°C. After a 1-week acclimatization period, the 8-week-old animals were divided into two groups. The MASH Group was fed with an HFD/methionine-choline-deficient (MCD) diet for 4 weeks to induce MASH, while the Control Group (db/m) received a methionine-choline-sufficient (MCS) diet for 4 weeks. All procedures involving animals were approved by the Institutional Animal Care and Use Committee (IACUC) of Harbin Medical University (Harbin, Heilongjiang Province, China), following the ethical guidelines for the care and use of laboratory animals. At the end of the treatment period, the mice were sacrificed under anesthesia using $CO_2$ inhalation, followed by cervical dislocation. Blood samples were collected by cardiac puncture under aseptic conditions and centrifuged at 3,000 rpm for 15 minutes at 4°C, and serum samples were collected. Liver tissues were excised for subsequent processing and collected into sterile frozen test tubes. In addition, the intestinal contents of three mice were placed in each test tube and immediately frozen in liquid nitrogen. All samples were stored at −80°C until further analysis.

### Transcriptome analysis

In the fourth week of modeling, liver samples were collected from the MASH Group and the Control Group (*n* = 3 per group) for sequencing. Total RNA was extracted from the tissue using TRIzol Reagent, quantified via ND-2000 (NanoDrop Technologies). Only high-quality RNA samples (OD260/280 = 1.8–2.2, OD260/230 ≥ 2.0, RQN ≥6.5, 28S:18S ≥ 1.0, >1 µg) were used for sequencing library construction. Shanghai Majorbio Bio-pharm Biotechnology Co., Ltd. (Shanghai, China) performed RNA purification, reverse transcription, library construction, and sequencing according to the manufacturer's instructions (Illumina, San Diego, CA). In short, Majorbio Bio-pharm Technology Co., Ltd constructed strand-specific libraries: 1 µg RNA underwent polyA selection, fragmentation, and double-stranded cDNA synthesis (SuperScript kit). Libraries were prepared using the Illumina Stranded mRNA Prep Ligation kit, size-selected (~300 bp, 2% agarose), and PCR-amplified (15 cycles). Paired-end RNA sequencing libraries were sequenced with the Illumina NovaSeq 6000 (2 × 150 base pair read length), following standard protocols from Majorbio Bio-Pharm Technology Co. Ltd. (Shanghai, China). The raw reads were trimmed and quality controlled by fastp with default parameters (55). Then clean reads were separately aligned to the reference genome with orientation mode using HISAT2 software (56). Employing a reference-based approach (57), the mapped reads were assembled using StringTie (https://ccb.jhu.edu/software/stringtie/index.shtml?tZexample); and gene abundances were quantified with the transcripts per million reads (TPM) method. RSEM was used to quantify gene abundances (58). Differential gene expression analysis was performed on the Majorbio Cloud Platform (https://www.majorbio.com/) (59) using DESeq2 (Wald test, *P* < 0.05), with batch effects adjusted by incorporating technical covariates into the design matrix (60). Significantly differentially expressed genes between MASH and Control Groups were identified with |log2 (fold change) | > 1 and *P* < 0.05. To explore the potential functions of DEGs, we

tested and visualized the statistical enrichment of DEGs in Gene Ontology (GO), the Kyoto Encyclopedia of Genes and Genomes (KEGG), and gene set enrichment analysis (GSEA) and also tested and visualized using the clusterProfiler R package (version 4.13.4) (61). In addition, the R package ggVolcano (version 0.0.2) was used to construct volcano plots, and the ggplot2 R package (version 3.5.1) was employed to construct bubble plots, lollipop plots, and histograms.

## Nontargeted serum metabolite profiling and analysis

Accurately weighed samples were homogenized in 2 mL tubes with 400 µL extraction solvent (acetonitrile:methanol = 1:1, vol/vol) containing 0.02 mg/mL L-2-chlorophenyla-lanine (internal standard). Vortex-mixed (30 s) and sonicated (5℃, 40 kHz, 30 min), followed by protein precipitation at −20℃ (30 min) and centrifugation (13,000 × $g$, 15 min, 4℃). The supernatant was dried under nitrogen, reconstituted in 100 µL acetonitrile:water (1:1, vol/vol), re-sonicated (5℃, 5 min), and centrifuged (13,000 × $g$, 10 min, 4℃). Quality control (QC) samples (pooled equal extracts) were inserted every 5–15 experimental samples. The injection and detection protocols for the QC samples were matched with those of the experimental samples. UHPLC-MS/MS analysis was performed on a Thermo UHPLC-Q Exactive HF-X system. Raw data were processed via Progenesis QI for peak extraction, calibration, and removal of internal standards/arti-facts (noise, column bleed). Metabolites features peak were identified and matched by searching reliable biochemical databases such as the HMDB (http://www.hmdb.ca/), Metlin (https://metlin.scripps.edu/), and Majorbio Database. Finally, MS data with a mass accuracy $<1.0 \times 10^{-5}$ were retained. Peak intensities from mass spectra were normalized using total ion count (TIC) normalization, followed by log transformation (lg). Metabo-lites with a relative standard deviation (RSD) >30% in the QC samples were removed to generate the final data set for subsequent analyses (62). Multivariate analyses (PCA, PLS-DA, OPLS-DA) were performed using the ropls package (v1.6.2) on the Majorbio Cloud Platform (https://cloud.majorbio.com) (59). The metabolites with VIP >1, $P < 0.05$ (MASH vs Control Group) were determined as significantly different metabolites based on the variable importance in the projection (VIP) obtained by the OPLS-DA model and the $P$-value generated by the Student's $t$-test. Differential metabolites were mapped into their biochemical pathways via metabolic enrichment and pathway analysis using MetaboAnalyst 6.0 (63), while R (R version 4.4.1) (64) was used to generate volcano plots, pie charts, and heatmaps.

## Gut microbiota statistics and bioinformatics analysis

High-throughput sequencing of full-length 16S rRNA was used to analyze the micro-biota composition in faecal samples ($n = 3$). Total DNA was extracted using FastPure Stool DNA Kit (MJYH, Shanghai, China), quantified with NanoDrop2000. The V3-V4 region was amplified with primers 338F (5′-AGCAG-3′) and 806R (5′-ACTCCTACGGGAG GC-GGACTACHVGGGTWTCTAAT-3′) (65) in 20 µL reactions: 4 µL 5× FastPfu buffer, 2 µL 2.5 mM dNTPs, 0.8 µL primers (5 µM), 0.4 µL FastPfu polymerase, 10 ng DNA. PCR conditions: 95℃/3 min; 27 cycles (95℃/30 s, 55℃/30 s, 72℃/45 s); 72℃/10 min. Triplicate amplicons were gel-purified (2% agarose), quantified (Synergy HTX), and sequenced on Illumina MiSeq PE300 at Majorbio Bio-Pharm Technology Co. Ltd. (Shanghai, China). Raw FASTQ files were de-multiplexed using an in-house Perl script, quality-filtered using fastp (v0.19.6) (55), based on a 50 bp sliding window (average quality score < 20, mini-mum read length 50 bp) and ambiguous character removal, and merged using FLASH (v1.2.11) (66) with a minimum overlap of 10 bp and a maximum mismatch ratio of 0.2. After barcode and primer-based sample identification and orientation correction (exact barcode matching, ≤2 nucleotide mismatches in primer matching), sequences were clustered into OTUs at 97% similarity using Usearch 11 (67, 68). Chloroplast sequences were manually removed. Samples were rarefed to 20,000 sequences, yielding an average Good's coverage of 99.09%. OTU taxonomy was assigned using RDP Classifier (v2.13) (69) against the Silva138.1/16s_bacteria database (confidence threshold 0.7). Bioinformatic

analysis of the gut microbiota was carried out using the Majorbio Cloud platform (https://cloud.majorbio.com). The Shannon and Simpson indices were calculated in Mothur (v1.30.2) (70). The Wilcoxon rank-sum test was used to analyze the intergroup differences in alpha diversity. Similarity among microbial communities was assessed via principal component analysis (PCA) based on Bray-Curtis dissimilarity, calculated using the Vegan v2.4-3 package. The linear discriminant analysis (LDA) effect size (LEfSe) (71) (http://huttenhower.sph.harvard.edu/LEfSe) was performed to identify the significantly abundant taxa (phylum to genera) of bacteria among the different groups (LDA score > 2, $P$ < 0.05). PICRUSt (http://picrust.github.io/picrust/) was used for KEGG function predictions. The entire analysis process adhered to the protocols of PICRUSt2 R (R version 4.4.1) (72). Metabolite origin and functional analysis were performed using MetOrigin (v2.0) (73) in Simple MetOrigin Analysis (SMOA) mode. SMOA then provides origin analysis to identify metabolite origins and metabolic function analysis, requiring a list of metabolites with KEGG or HMDB IDs.

## Correlation analysis

The Selected DEGs, DMs, and differential genera are shown in Table S4. We performed the Spearman correlation analysis using the "corr.test" function of the R package "psych" (version 2.6-8) to analyze the interactions among DEGs, DMs, and differential genera. The associations between genes and microbes with an absolute correlation coefficient >0.81 and $P$ < 0.05 are regarded as significant interactions. The Mantel test analysis was employed as a correlation test method, determining the correlation between two matrices of distance measures and helping determine whether sample distances in one matrix are correlated with sample distances in another matrix. The Mantel test analysis was used to determine the correlation between DEGs in the transcriptome and DMs in the metabolome (metabolite distance matrix: Euclidean distance; gut microbiota distance matrix: Euclidean distance; gene expression distance matrix: Bray-Curtis distance) using the "mantel_test" function of the R package "ggcor" (version 0.9.8.1). The R package "circlize" (version 0.4.16) was used for correlation circle plots, and the R package "ggplot2" (version 3.5.1) was used for correlation dotplots.

## ACKNOWLEDGMENTS

This work was supported by grants from the National Natural Science Foundation of China (No. 82170268), Heilongjiang Province key research and development plan project (No. 2022ZX06C17), and the Outstanding Medical Fund for Young Scholars of the First Affiliated Hospital of Harbin Medical University (No. HYD2020YQ0014).

Z.L. and L.Y., Methodology and Writing—original draft; L.Y., Y.B., and Y.C., Data curation, Software, and Writing—original draft; M.S. and Z.L., Methodology; X.L. and X.Z., Data curation and Visualization; Y.J., Project administration and Conceptualization; C.W.: Project administration, Conceptualization, Resources, Data curation and Supervision.

## AUTHOR AFFILIATIONS

[1]Department of General Surgery, Key Laboratory of Hepatosplenic Surgery, Ministry of Education, The First Affiliated Hospital of Harbin Medical University, Harbin, Heilongjiang, China
[2]Department of Pathophysiology, Harbin Medical University, Harbin, Heilongjiang, China
[3]Department of Oncology, The Second Affiliated Hospital of Harbin Medical University, Harbin, Heilongjiang, China

## AUTHOR ORCIDs

Ligen Yu http://orcid.org/0009-0009-2698-2421
Ye Jin https://orcid.org/0009-0003-6646-2240
Can Wei http://orcid.org/0000-0003-4291-7653

## AUTHOR CONTRIBUTIONS

Zhaoyang Lu, Methodology, Writing – original draft | Ligen Yu, Data curation, Methodology, Software, Writing – original draft | Yun Bai, Data curation, Software, Writing – original draft | Yifeng Cui, Data curation, Software, Writing – original draft | Meixin Shi, Methodology | Zhitao Li, Methodology | Xiaoxue Li, Data curation, Visualization | Xin Zhong, Data curation, Visualization | Ye Jin, Conceptualization, Project administration | Can Wei, Conceptualization, Data curation, Project administration, Resources, Supervision

## DATA AVAILABILITY

All the information supporting the conclusions is provided within the paper. The raw reads of transcriptomics sequencing were deposited into the NCBI Sequence Read Archive (SRA) database (BioProject Number: PRJNA1201862). The raw reads of 16S rRNA sequencing were deposited into the NCBI Sequence Read Archive (SRA) database (Bioproject Number: PRJNA1247707). Processed metabolome data, including detailed information about metabolites identified and their respective abundances, are presented in Table S5.

## ETHICS APPROVAL

The mice were treated following the Guide for the Care and Use of Laboratory Animals as adopted and promulgated by the US NIH. All treatment protocols were approved by the Institutional Animal Care and Use Committee at the Experimental Animal Center of Harbin Medical University (Harbin, Heilongjiang, China).

## ADDITIONAL FILES

The following material is available online.

### Supplemental Material

**Supplemental material (mSystems00518-25-s0001.docx).** Tables S1 and S2; Fig. S1 to S3.
**Approval (mSystems00518-25-s0002.pdf).** IACUC approval document.
**Table S3 (mSystems00518-25-s0003.xlsx).** MetOrigin-based annotation of 267 metabolites.
**Table S4 (mSystems00518-25-s0004.xlsx).** Overlapping metabolic pathways between PICRUSt2 and MESA.
**Table S5 (mSystems00518-25-s0005.xlsx).** Metabolite information overview.

### Open Peer Review

**PEER REVIEW HISTORY (review-history.pdf).** An accounting of the reviewer comments and feedback.

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
