## [Reviewer comments · mSystems]

The Potential Mechanisms of Reciprocal Regulation of Gut microbiota-liver immune Signalling in Metabolic Dysfunction-Associated Steatohepatitis Revealed in Multi-Omics Analysis

Zhaoyang Lu, Iigen Yu, Yun Bai, Yifeng Cui, Meixin Shi, Zhitao Li, Xiaoxue Li, Xin Zhong, Ye Jin, and Can Wei

Corresponding Author(s): Can Wei, Harbin Medical University

Review Timeline:

Submission Date:	April 14, 2025
Editorial Decision:	May 8, 2025
Revision Received:	May 9, 2025
Accepted:	May 13, 2025

Editor: Jack Gilbert

Reviewer(s): Disclosure of reviewer identity is with reference to reviewer comments included in decision letter(s). The following individuals involved in review of your submission have agreed to reveal their identity: Hongzhu Li (Reviewer #1); Yuehong Wang (Reviewer #2)

Transaction Report:

DOI: <https://doi.org/10.1128/msystems.00518-25>

Re: mSystems00518-25 (**The Potential Mechanisms of Reciprocal Regulation of Gut microbiota-liver immune Signalling in Metabolic Dysfunction-Associated Steatohepatitis Revealed in Multi-Omics Analysis**)

Dear Dr. Can Wei:

Revision Guidelines

Sincerely,
Jack Gilbert
Editor
mSystems

Reviewer #1 (Comments for the Author):

No.

Reviewer #2 (Comments for the Author):

The authors well addressed each of my comments. I have no further comments.

Reviewer #3 (Comments for the Author):

The authors addressed my major concerns from the last review period. However, there a few minor changes that would improve this manuscript.

Minor comments:

Lines 46 and 353: Remove "genus" after *Enterococcus* and *Romboutsia*. This is redundant after stating "microbiota genera"

Lines 53 and 54: Remove either "can" or drop the "s" from "provides"

Line 66: Remove comma after clarify.

Line 94: Remove commas from this sentence after lacking and choline.

Line 113: Remove "By applying a threshold of $|\log_2FC| \geq 1$ and $p < 0.05$ " - This can remain in methods

Line 133: Remove "s" from "roles"

Line 175: Remove "pinpointed"

Line 233: "Likely, the 619 differentially expressed genes (DEGs) identified by RNA-seq in this study are likely critical to the pathogenesis of MASH, and the up-regulation and down-regulation of these genes are likely involved in the occurrence, progression, and persistence of the disease."

- To say that all 619 DEGs are likely involved in disease development is to overstate the findings. You only have a sample size of 6 (3 per group), but even if the sample size were much larger, there are a number of reasons that DEGs might be identified yet be artifactual, including an unidentified co-variate or the limitations of the database used for identification (there's quite a lot of variation between different software packages that conduct the same type of analyses). I would walk this back a little and say that they are "potentially involved in" or change the sentence to say "The up-regulation and down-regulation of the 619 DEGs are potentially involved in the occurrence, progression, and persistence of the disease." - This will also remove the redundancy from the first and last sentence of the paragraph.

Lines 283 - 285: Families and phyla are not italicized, and Firmicutes should be capitalized.

Line 292: This last sentence doesn't make any sense.

Line 297: Reference after "humans"

General: Still lots of spacing/period/comma issues, but I imagine that can be fixed at the proofing stage.

General: There is quite a bit of info in the discussion that should be in the intro. For example, "The bile acid enterohepatic circulation between the gut and liver plays a critical role in maintaining host metabolic homeostasis (35). As the co-metabolites derived from host synthesis and microbial modification, bile acids can directly regulate the metabolic-dysfunction-associated steatohepatitis (MASH) through activating nuclear receptors including farnesoid X receptor (FXR) and Takeda G protein-coupled receptor 5 (TGR5), while indirectly shaping gut microbiota composition via modulating the intestinal epithelial microenvironment (36-39). Clinical studies have demonstrated that during the progression of metabolic dysfunction-associated fatty liver disease (MAFLD), bile acid levels increase (40), and this phenomenon is closely linked to the bile acid dysregulation induced by obesity and type 2 diabetes (T2D) (41, 42). Such metabolic disturbances drive structural remodelling of the gut microbiota, characterized by decreased Bacteroidetes and increased Firmicutes abundance, a pattern consistently observed in high-fat diet-fed db/db mouse models."

General: There are also a lot of Results in the Discussion section: For example, "Using the Metorigin database for traceability of gut-derived metabolites and the PICRUST2 analysis for predicting microbiota functions, we found the same KEGG pathways between them: glycine, serine and threonine metabolism, Glycerophospholipid metabolism, Pentose and glucuronate interconversions pathway, etc. (45) Among the metabolites of these pathways, based on the Mantel-test analysis and Spearman correlation analysis, L-glutamine, Isocitric acid, Putrescine, Pyroglutamic acid, Rhamnose may be potential messenger molecules for the interactions between the gut-liver axis. Aligned with this, metabolites enriched in the 18 shared KEGG pathways were further analyzed using the Mantel test and Spearman correlation analyses to explore their associations with transcriptomics and differential genera. Among them, L-glutamine, Isocitric acid, Putrescine, Pyroglutamic acid, and Rhamnose exhibited stronger correlations with metabolic genes associated with these pathways in the multi-omics analyses. Our findings

demonstrate that the gut microbiota and its their metabolites, particularly putrescine and rhamnose, are implicated in the progression of metabolic dysfunction-associated steatohepatitis (MASH)."

- This entire section can be cut down to: "We found evidence that L-glutamine, Isocitric acid, Putrescine, Pyroglutamic acid, and Rhamnose may act as potential messenger molecules in the gut-liver axis, and these findings suggest that gut-associated microbiota and metabolites, particularly Putrescine and Rhamnose, are implicated in the progression MASH."

Dear editor,

Please see below for my comments on the review of “The Potential Mechanisms of Reciprocal Regulation of Gut microbiota-liver immune Signalling in Metabolic Dysfunction-Associated Steatohepatitis Revealed in Multi-Omics Analysis.”

The authors addressed my major concerns from the last review period. However, there are a few minor changes that would improve this manuscript.

Minor:

Lines 46 and 353: Remove “genus” after *Enterococcus* and *Romboutsia*. This is redundant after stating “microbiota genera”

Lines 53 and 54: Remove either “can” or drop the “s” from “provides”

Line 66: Remove comma after clarify.

Line 94: Remove commas from this sentence after lacking and choline.

Line 113: Remove “By applying a threshold of $|\log_2FC| \geq 1$ and $p < 0.05$ ” – This can remain in methods

Line 133: Remove “s” from “roles”

Line 175: Remove “pinpointed”

Line 233: “Likely, the 619 differentially expressed genes (DEGs) identified by RNA-seq in this study are likely critical to the pathogenesis of MASH, and the up-regulation and down-regulation of these genes are likely involved in the occurrence, progression, and persistence of the disease.”

- To say that all 619 DEGs are likely involved in disease development is to overstate the findings. You only have a sample size of 6 (3 per group), but even if the sample size were much larger, there are a number of reasons that DEGs might be identified yet be artifactual, including an unidentified co-variate or the limitations of the database used for identification (there’s quite a lot of variation between different software packages that conduct the same type of analyses). I would walk this back a little and say that they are “potentially involved in” or change the sentence to say “The up-regulation and down-regulation of the 619 DEGs are potentially involved in the occurrence, progression, and persistence of the disease.” – This will also remove the redundancy from the first and last sentence of the paragraph.

Lines 283 – 285: Families and phyla are not italicized, and Firmicutes should be capitalized.

Line 292: This last sentence doesn’t make any sense.

Line 297: Reference after “humans”

General: Still lots of spacing/period/comma issues, but I imagine that can be fixed at the proofing stage.

General: There is quite a bit of info in the discussion that should be in the intro. For example, “The bile acid enterohepatic circulation between the gut and liver plays a critical role in maintaining host metabolic homeostasis (35). As the co-metabolites derived from host synthesis and microbial modification, bile acids can directly regulate the metabolic-dysfunction-associated steatohepatitis (MASH) through activating nuclear receptors including farnesoid X receptor (FXR) and Takeda G protein-coupled receptor 5 (TGR5), while indirectly shaping gut microbiota composition via modulating the intestinal epithelial microenvironment (36-39). Clinical studies have demonstrated that during the progression of metabolic dysfunction-associated fatty liver disease (MAFLD), bile acid levels increase (40), and this phenomenon is closely linked to the bile acid dysregulation induced by obesity- and type 2 diabetes (T2D) (41, 42). Such metabolic disturbances drive structural remodelling of the gut microbiota, characterized by decreased Bacteroidetes and increased Firmicutes abundance, a pattern consistently observed in high-fat diet-fed db/db mouse models.”

General: There are also a lot of Results in the Discussion section: For example, “Using the Metorigin database for traceability of gut-derived metabolites and the PICRUST2 analysis for predicting microbiota functions, we found the same KEGG pathways between them: glycine, serine and threonine metabolism, Glycerophospholipid metabolism, Pentose and glucuronate interconversions pathway, etc. (45) Among the metabolites of these pathways, based on the Mantel-test analysis and Spearman correlation analysis, L-glutamine, Isocitric acid, Putrescine, Pyroglutamic acid, Rhamnose may be potential messenger molecules for the interactions between the gut-liver axis. Aligned with this, metabolites enriched in the 18 shared KEGG pathways were further analyzed using the Mantel test and Spearman correlation analyses to explore their associations with transcriptomics and differential genera. Among them, L-glutamine, Isocitric acid, Putrescine, Pyroglutamic acid, and Rhamnose exhibited stronger correlations with metabolic genes associated with these pathways in the multi-omics analyses. Our findings demonstrate that the gut microbiota and its their metabolites, particularly putrescine and rhamnose, are implicated in the progression of metabolic dysfunction-associated steatohepatitis (MASH).”

- This entire section can be cut down to: “We found evidence that L-glutamine, Isocitric acid, Putrescine, Pyroglutamic acid, and Rhamnose may act as potential messenger molecules in the gut-liver axis, and these findings suggest that gut-associated microbiota and metabolites, particularly Putrescine and Rhamnose, are implicated in the progression MASH.”

Re: mSystems00518-25R1 (**The Potential Mechanisms of Reciprocal Regulation of Gut microbiota-liver immune Signalling in Metabolic Dysfunction-Associated Steatohepatitis Revealed in Multi-Omics Analysis**)

Dear Dr. Can Wei:

Your manuscript has been accepted, and I am forwarding it to the ASM production staff for publication. Your paper will first be checked to make sure all elements meet the technical requirements. ASM staff will contact you if anything needs to be revised before copyediting and production can begin. Otherwise, you will be notified when your proofs are ready to be viewed.

Sincerely,
Jack Gilbert
Editor
mSystems